# Consumers' Willingness to Purchase Imported Cherries towards Sustainable Market: Evidence from the Republic of Korea



**Seongmin Shin** [1,2] and **Seongtae Ji** [2,3,*]

1   Centre for International Forestry Research (CIFOR), Jalan CIFOR, Situ Gede, Bogor 16115, Indonesia; seongmin.shin@cgiar.org
2   Graduate School of International Agricultural Technology, Seoul National University, Seoul 08826, Korea
3   Institutes of Green Bio Science and Technology, Seoul National University, Seoul 08826, Korea
*   Correspondence: dongsimjst@snu.ac.kr; Tel.: +82-33-339-5707

**Abstract:** Globalization has led diverse stakeholders to join the market and has resulted in corporate and product diversification; however, some markets remain monopolized by a few countries owing to "shadow trade barriers" influencing willingness to purchase. The Korean cherry market has grown rapidly since 2000 but is monopolized by U.S. cherries, which makes the market unsustainable; however, Uzbekistan cherries are 1.75-times less expensive than U.S. cherries. We examined the potential of Uzbekistan cherries to replace U.S. cherries as cherries are imported only from these countries during the spring–summer season. After collecting data through a web survey, we conducted logistic regression analyses to investigate what specific factors affect Korean consumers' willingness to purchase Uzbekistan cherries over U.S. cherries: price perception, brand familiarity, perceived risk, and country of origin. Results showed that the more price awareness (price perception), experience (brand familiarity), and higher confidence of safety (perceived risk) that consumers had, the more they were willing to purchase Uzbekistan cherries. Consumers who checked the country of origin were less likely to purchase Uzbekistan cherries. The results provide useful information for exporters, importers, researchers, decision-makers, and policymakers concerning the utilization of products for sustainability in a monopolized market.

**Keywords:** willingness to purchase; fruit; cherry; country of origin; marketing; monopolization; consumers' preference; market sustainability

## 1. Introduction

Globalization has reduced technical barriers to trade through negotiations and encouraged stakeholders to join the global market, which in turn has led to heightened competition in national and global markets [1]. Overall, globalization has resulted in corporate and product diversification; however, some markets remain monopolized by a few countries or large companies [2]. The regulated markets have had several negative impacts on sustainability in the market, such as higher prices in spite of cost advantages, allocative inefficiency, production inefficiency, supernormal profit, lack of choice, and less innovation than in a competitive market [3]. The main reasons why monopolies emerge are the pre-existing monopoly power of developed countries and "shadow trade barriers" that influence consumers' willingness to purchase (WTP) [2]. To enhance food security and minimize the negative impacts of monopolies against sustainability, it is critical to draw up solutions to reduce import reliance and encourage diversification in import sources [4]. Although developing countries have tried to break into pre-occupied markets with less expensive products, it is becoming increasingly difficult to benefit equitably from globalization owing to trade barriers [5]. Developing countries also experience a lack of capacity in meeting quality demands in global markets [6].

Cherries are an imported fruit in South Korea, which is witnessing an increasing trend of cherry imports [7]. The volume of cherry imports was only about 5000 tons in 2011;

however, it increased to 18,000 tons in 2018, with an average annual growth rate of 20.2% (Figure 1). Cherry imports are also expected to increase in the future, as Korean consumers have a strong preference for them. Korea's main import partner for cherries is the U.S., and the proportion of U.S. cherries (90%) is far higher than that of cherries from other small partners such as Uzbekistan and Chile [7]. In other words, the U.S. has a monopoly over the Korean cherry market, although the unit price of U.S. cherries is almost twice as high as the price of Uzbekistan cherries (Table 1). Cherry imports are concentrated in the period from May to August (Figure 2). This is also a season when U.S. cherries are imported intensively. The only alternatives are Uzbekistan cherries in the same importing season; however, their imports are relatively small (less than 1%). Other options from Chile, Australia, and New Zealand are mainly imported from November to February but do not compete with U.S. cherries because of the difference in the importing season (Figure 2).

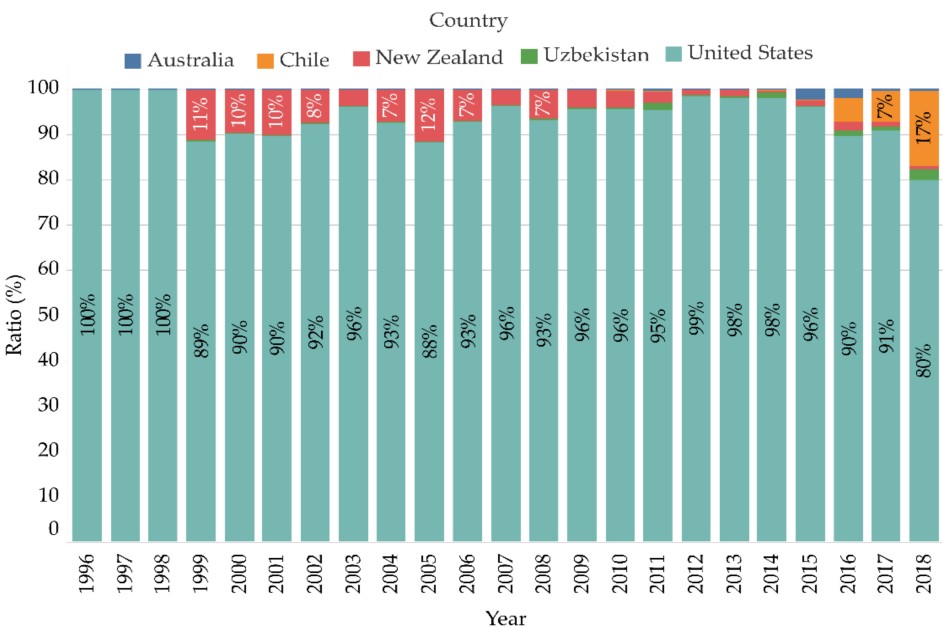

**Figure 1.** Korea's cherry import trend. Source: Drawing by the author based on data from Trade Statistics Service.

|  | Oct | Nov | Dec | Jan | Feb | Mar | Apr | May | Jun | Jul | Aug | Sep |
|---|---|---|---|---|---|---|---|---|---|---|---|---|
| Chile |  |  |  |  |  |  |  |  |  |  |  |  |
| Australia |  |  |  |  |  |  |  |  |  |  |  |  |
| New Zealand |  |  |  |  |  |  |  |  |  |  |  |  |
| United States |  |  |  |  |  |  |  |  |  |  |  |  |
| Uzbekistan |  |  |  |  |  |  |  |  |  |  |  |  |
| Ratio (%) | 0.0 | 0.6 | 2.3 | 5.7 | 2.0 | 0.0 | 0.5 | 18.8 | 34.8 | 29.0 | 6.3 | 0.0 |

**Figure 2.** Monthly cherry import status of Korea. Source: Drawing by the author based on data from Trade Statistics Service. Note: The monthly proportion was calculated based on the three-year import statistics from October 2015 to September 2018. The brightness of shading indicates the difference in the proportion of monthly imports.

**Table 1.** Cherry import unit price by country.

| Year | Average | Uzbekistan | United States | Chile | Australia | New Zealand |
|------|---------|------------|---------------|-------|-----------|-------------|
| 2010 | 9581 | 3916 | 8516 | - | 13,214 | 12,679 |
| 2011 | 9866 | 4300 | 9598 | - | 12,882 | 12,684 |
| 2012 | 10,221 | 3900 | 8679 | - | 14,517 | 13,789 |
| 2013 | 10,126 | 3400 | 9793 | - | 12,558 | 14,754 |
| 2014 | 10,511 | 3739 | 9409 | - | 13,039 | 15,859 |
| 2015 | 9764 | 4162 | 9925 | - | 12,438 | 12,529 |
| 2016 | 9257 | 3600 | 8876 | 11,826 | 10,663 | 11,323 |
| 2017 | 9055 | 2757 | 9061 | 9658 | 11,631 | 12,168 |
| 2018 | 8714 | 5213 | 9140 | 8941 | 9143 | 11,134 |

Unit: USD ($). Source: writing by the author based on data from source.

Although U.S. cherries monopolize the Korean cherry market, Uzbekistan cherries have high export potential due to price competitiveness, pre-cooling, and packaging technology [8]. By upgrading the value chain encompassing production, processing, distribution, and exports, Uzbekistan cherries can be expected to expand their market share in the Korean cherry market and possibly replace U.S. cherries. However, it is difficult to guarantee whether Uzbekistan cherry exports to Korea will increase, because of the uncertainty regarding Korean consumers' preferences and WTP for Uzbekistan cherries. Therefore, it is necessary to examine the factors that influence Korean consumers' purchase of cherries in advance. Such factors, including empirical purchases, origin, pricing, and quality, create a certain brand image that affects consumers' WTP. Therefore, it is necessary to analyze consumers' personalities and use appropriate marketing strategies to establish long-term competitiveness in the market.

We thus examined whether price perception, brand familiarity, perceived risk, and country of origin influence consumers' WTP for imported cherries and determined the potential of cherries imported from developing countries, especially Uzbekistan, in the Korean cherry market, which is currently monopolized by the U.S. cherry imports. We investigated the factors affecting Korean consumers' WTP for Uzbekistan cherries, assuming a quality improvement in Uzbekistan cherries as compared to U.S. cherries.

## 2. Materials and Methods

Consumers' WTP has been widely researched, especially in terms of consumers' psychology; however, few studies have focused on the WTP for imported fruits. Previous research has explored diverse predictors of WTP, such as product information [9,10], price perception [11–13], brand familiarity [13,14], perceived risk [15–18], and country of origin [14,19]. Concerning the WTP for imported fruits, Kathuria and Singh [10] used nutrition, price, perceived risk, country of origin, labeling, and brand image as important variables affecting the WTP for imported fruits in India. Considering the distinct characteristics of cherries and the general attributes of WTP, we set price perception, brand familiarity, perceived risk, and country of origin as influential factors, to model hypotheses based on the utility theory. We postulated a conceptual model of the WTP for cherries in Korea.

### 2.1. Price Perception (Value for Money)

Consumers purchase any product or service while considering its utility [20]. In other words, the more value compared to the price that products have, the greater is the demand for these products. This is the utility maximization rule [21]. The utility is the subjective state of satisfaction that consumers obtain when consuming goods for a certain period [22]. As price perception depends on the subjective perspective of consumers, such as utility, perception varies with the purchasing power or preferences of individuals. Generally, value is evaluated by comparing the perceived price and quality of products [23]. Perceived price also affects consumers' choice of brand [12] and perception of product quality [24]. For

example, Kim and Ha [25] found that price fisheries have a significant impact on consumer satisfaction and WTP.

Moreover, consumers consider the relative price to be more important than the actual cost [24]. Consumers value products by comparing their prices with those of other companies or from past purchasing experience. Hence, a favorable perceived price facilitates utility and WTP. Consequently, we posited that Korean consumers would set a low value on cherries as they perceive the market price as high. They would thus be more likely to choose less-expensive products with a low cherry value rather than considering high quality or other properties. Consequently, during the competition between Uzbekistan cherries and U.S. cherries in the Korean imported cherry market, it is highly feasible that consumers would prefer Uzbekistan cherries if the quality is increasing while maintaining lower prices than the U.S. cherries. Those who purchase U.S. cherries but have lower satisfaction with the current cherry price may switch to Uzbekistan cherries.

**Hypothesis 1 (H1).** *Consumers with lower (vs. higher) satisfaction with the current cherry price will be more willing to purchase less-expensive cherries from Uzbekistan.*

In addition, we assumed that those who place great importance on price would have a low valuation of product quality. Korean consumers would be more favorable to Uzbekistan cherries than U.S. cherries when the quality of both cherries is equal but when Uzbekistan ones are less expensive.

**Hypothesis 1-1 (H1-1).** *Consumers who value price more than quality will be more willing to purchase cherries from Uzbekistan at lower prices as compared to the U.S. cherries.*

### 2.2. Brand Familiarity (Purchasing Experience)

Brand familiarity means to have a positive impression about the consumption experience of a specific brand [26], and familiarity is based on the level of direct and indirect experience [27]. The concept of brand familiarity is used to raise brand recognition using diverse communication methods, which is a marketing strategy to make consumers purchase familiar products [13]. Brand familiarity is considerably related to brand awareness, brand image, brand trust, and eventually decision-making [28].

Kwag and Ryu [29] investigated consumers' brand preferences to check product information when purchasing. Their results showed that people tend to choose a familiar brand of the same quality as the product. Generally, purchasing experience and advertising exposure have a positive impact on brand familiarity. The country of origin replaces brand familiarity when no specific brand stands out (e.g., fruits). In this case, the purchasing experience highly affects brand familiarity through the verification of quality. As mentioned above, only the U.S. cherry has a large market share in Korea, which means that the brand familiarity of Uzbekistan cherries is relatively low in the Korean cherry market. Therefore, we suppose that the brand familiarity of those who have experienced Uzbekistan cherries is higher than that of those who do not.

**Hypothesis 2 (H2).** *Consumers with higher (vs. lower) brand familiarity (purchasing and eating experience) will be more willing to purchase cherries from that brand.*

### 2.3. Perceived Risk (Safety)

Consumers try to minimize the risk factors in the purchase decision-making process. Of course, although risk factors exist in this process, these factors will not influence the purchasing decision-making process if consumers are not aware of them [30], which is an unknown and uncertain risk in purchasing decisions.

Perceived risk is determined by consumers' supervision [31], and its contents include financial, functional, physical, psychological, and social risks [32]. Safety among the perceived risks of food purchases had the most significant impact in that previous study [32]. Food safety is a severe risk factor because it is directly associated with individuals' health

or even mortality. There are financial and functional risks from purchasing, but they are secondary risk factors compared to physical risks.

To overcome perceived risks, consumers should actively explore information and seek countermeasures, which affect their attitude and WTP [33]. When choosing imported agricultural products such as cherries, consumers assess the risks of the products based on relevant information that the purchasers have and reflect them in their purchasing behaviors. It is difficult to ensure accuracy because perceived risk depends on the information held by individuals. Direct and indirect information, including purchasing experience and the image of the exporting country, will affect final decision-making.

Consumers assess the safety of cherries imported from each country, which also depends heavily on their purchasing experience and the image of the exporting country. The higher the assessment of safety, the greater the consumers' WTP. In contrast, perceived risk has a more pronounced impact if assessed to be relatively less secure.

**Hypothesis 3 (H3).** *Consumers with less (vs. more) perceived risk of cherry origin (U.S. or Uzbekistan) will be more willing to purchase cherries from that country.*

### 2.4. The Country of Origin (National Preference)

The country of origin has a significant impact on purchasing decisions because it represents the quality of the product as well as its value and safety [34]. In other words, the country of origin serves as a key factor in making a purchase decision because it implies additional information presented earlier, such as price-to-value, perceived risk, and brand familiarity. The country of origin also plays an essential role in determining the price [35] and perceive hazards, including food safety, so that consumers can assess products differently according to the country of origin [36]. In addition, the halo effect of origin has a positive impact on national image and brand familiarity [34]. Even if brand familiarity is low, the stability and understanding of the country of origin may increase the purchase rate [37].

The origin of raw materials is sometimes recognized as the origin of such food, especially for processed foods [38], which is also related to the country of origin effect. The recent opening of agri-food trade and the multi-nationalization of food-related companies have made the country of origin a critical consideration for consumers' food purchases [38,39]. If the country of origin is a developed country, it is more likely to get a more positive evaluation than developing countries regarding product quality, safety, and value. Therefore, identifying the country of origin can be understood as a preference for developed countries' products like Americans, among other countries of origin. Hence, recognition of origin is a self-rescue measure used by consumers to guard against the monopolized market, where relatively low-quality goods are distributed.

**Hypothesis 4 (H4).** *Consumers who check the country of origin will be more willing to purchase cherries from advanced countries as compared to their counterparts.*

### 3. Design and Data

### 3.1. Survey Design and Data

Data were collected through EMBRAIN, a survey company, from July 9 to July 23, 2019. Participants were adult women, aged 20 years or older, and living in Seoul, Incheon, or Gyeonggi-do, Korea ($N$ = 584). Participants were asked, "If the quality is the same, which would you prefer, American or Uzbekistan cherries?" (Table 2). Among them, 272 participants (46.6%) said they would purchase U.S. cherries, 286 (49.0%) responded that they would make their purchase regardless of the origin, and 26 (4.4%) replied that they would purchase Uzbekistan cherries.

We aimed to determine whether consumers' intention to purchase cherries would change in the monopoly market occupied by the U.S. We examined consumers' preference between U.S. and Uzbekistan cherries since the only competitor over the U.S. is Uzbekistan

in the season when U.S. cherries are imported into the Korean cherry market. We excluded 108 respondents who had no experience purchasing cherries over the past three years. Finally, 476 people were asked, "If the quality (newness, sugar content) of Uzbekistan cherries is the same as that of the U.S., would you be willing to purchase them?"

**Table 2.** Descriptive tabulation of explanatory variables.

| Construct | Definition | Value |
|---|---|---|
| Willingness to purchase | Consumers' willingness to purchase Uzbekistan cherries if they meet the same quality as the U.S. cherries and are less expensive than the U.S. cherries | 0: Uzbekistan, 1: U.S. |
| Price perception | 1: The degree that consumers feel appropriate about the current price of cherries or not. | 1: very cheap, 2: affordable, 3: reasonable, 4: expensive, 5: highly expensive |
| | 2: The degree that consumers consider price important when purchasing cherries. | 0 points to 1 point |
| Brand familiarity | 1: Experience eating U.S. cherries. | 0: never tried, 1: tried U.S. cherries |
| | 2: Experience eating Uzbekistan cherries. | 0: never tried, 1: tried Uzbekistan cherries |
| Perceived Risk | 1: Whether consumers feel safe about foods imported from the U.S or not. | 1: highly risky, 2: risky, 3: moderate, 4: safe, 5: very safe |
| | 2: Whether consumers feel safe about foods imported from Uzbekistan or not. | 1: highly risky, 2: risky, 3: moderate, 4: safe, 5: very safe |
| Country of Origin | Whether consumers buy cherries with the awareness of country of origin, or not. | 0: no check, 1: checking country of origin |

### 3.2. Analysis

Participants' demographic data are displayed with frequency analysis in Table 3.

**Table 3.** Participants' demographic characteristics.

| Variable | Group | Coding Value | N | % | F ($p > z$) * |
|---|---|---|---|---|---|
| | 20–29 | 1 | 86 | 18.07 | 2.6 (0.036) |
| | 30–39 | 2 | 96 | 20.17 | |
| Age (years) | 40–49 | 3 | 96 | 20.17 | |
| | 50–59 | 4 | 101 | 21.22 | |
| | 60–69 | 5 | 97 | 20.38 | |
| Homemaker or other | Homemaker | 1 | 127 | 26.68 | 0.38 (0.538) |
| | Other | 0 | 349 | 73.32 | |
| Marital status | Married | 1 | 312 | 65.55 | 0.07 (0.791) |
| | Unmarried | 0 | 164 | 34.45 | |
| | 0 | 0 | 32 | 6.72 | 0.56 (0.728) |
| | 1 | 1 | 100 | 21.01 | |
| Number of family members | 2 | 2 | 126 | 26.47 | |
| | 3 | 3 | 167 | 35.08 | |
| | 4 | 4 | 47 | 9.87 | |
| | 5 | 5 | 4 | 0.84 | |
| | 1,000,000–3,000,000 won | Income level 1 | 104 | 21.85 | 2.62 (0.035) |
| | 3,000,001–5,000,000 won | Income level 2 | 169 | 35.50 | |
| Income | 5,000,001–7,000,000 won | Income level 3 | 126 | 26.47 | |
| | 7,000,001–9,000,000 won | Income level 4 | 50 | 10.50 | |
| | >9,000,001 won | Income level 5 | 27 | 5.67 | |
| | < 5 times | 1 | 150 | 31.51 | 0.69 (0.600) |
| | 5–9 times | 2 | 205 | 43.07 | |
| Number of all fruits purchases | 10–15 times | 3 | 86 | 18.07 | |
| | 16–19 times | 4 | 16 | 3.36 | |
| | ≥20 times | 5 | 19 | 3.99 | |
| | Supermarket | Place 1 | 268 | 56.30 | 1.30 (0.270) |
| | Small market | Place 2 | 59 | 12.39 | |
| Place to purchase fruits | Traditional market | Place 3 | 74 | 15.55 | |
| | Online market | Place 4 | 28 | 5.88 | |
| | Fruit store | Place 5 | 47 | 9.87 | |

* F and $p$ value from ANOVA and *t*-test.

In order to figure out WTP of the demographic characteristics, analysis of variance (ANOVA) and *t*-test were conducted (Table 3). In addition to these household characteristics, we assumed that various factors affected cherry purchases, such as price perception, brand familiarity, perceived risk, and country of origin. To determine ways to overcome the monopolized market, we examined the factors that significantly impact the WTP for imported cherries in the Korean market. We posited that the quality of Uzbekistan cherries has been improved to the level of the U.S., and the price is relatively low. We then analyzed the key factors that affect Korean consumers' WTP using logistic regression analyses.

In this analysis, the dependent variable, WTP, was dichotomous, that is, U.S. or Uzbekistan cherries. The WTP for U.S. cherries was coded as 1 while WTP for Uzbekistan cherries was coded as 0. A binary logistic regression analysis [40] was employed to model this dichotomous choice and investigate the influence of the dependent variables on participants' WTP for cherries. We defined the probability, $p$ (WTP = 1), as $p$. First, probabilities were transformed into odds, $p/(1-p)$, to determine p from the predictive variables [41]. Then, we returned the logarithm of the odds to obtain the following equation:

$$Z_i = \log(odds) = \ln\left(\frac{p}{1-p}\right) \tag{1}$$

where $Z_i$ is the log odds of respondents' WTP [41]. By using Equation (1), we solved $p$ as follows:

$$p = \frac{\exp(Z_i)}{1 - \exp(Z_i)} = \frac{\exp(\alpha + \beta_1 x_1 + \beta_2 x_2 + \cdots\cdots + \beta_n x_n)}{1 - \exp(\alpha + \beta_1 x_1 + \beta_2 x_2 + \cdots\cdots + \beta_n x_n)} \tag{2}$$

Finally, the following equation was used for the binary logistic regression analysis, as a function of the following predictors:

$$Z_i = \log(odds) = \ln\left(\frac{p}{1-p}\right) = \alpha + \beta_1 x_1 + \beta_2 x_2 + \cdots\cdots + \beta_n x_n + \varepsilon \tag{3}$$

where the intercept was a; the dependent variables, or predictors, were $x_1$ to $x_n$; the parameter coefficients were $\beta_1$ to $\beta_n$; and $\varepsilon$ was the error term [42]. The parameter coefficients of $\beta$ show the amount of increase in participants' WTP in response to a one-unit increase in the predictor variable while holding other variables constant [43]. Based on participants' responses, the independent variables were household characteristics, price perception (PP), brand familiarity (BF), perceived risk (PR), and country of origin (CO), as listed in Table 2. The final equation substituting $x_1$ $x_n$ with the predictors (Table 2) is as follows:

$$
\begin{aligned}
Z_i = \log(odds) \quad &= \ln\left(\frac{p}{1-p}\right) \\
&= \alpha + \beta_1 \text{Age} + \beta_2 \text{Homemaker} + \beta_3 \text{Marriage} \\
&+ \beta_4 \text{Number of Family Member} + \beta_5 \text{Income} \\
&+ \beta_6 \text{Number of Fruit Purchase} + \beta_7 \text{Place to buy} \\
&+ \beta_8 \text{Price Perception (PP)} + \beta_9 \text{Brand Familiarity (BF)} \\
&+ \beta_{10} \text{Perceived Risk (PR)} + \beta_{11} \text{Country of Origin (CO)} + \varepsilon
\end{aligned} \tag{4}
$$

The model in Equation (4) shows the significant factors of WTP with the parameter coefficients. For analysis, Stata 16.0. software [44] was used.

## 4. Empirical Results

Table 4 indicates the fitness of the estimation model and the coefficient estimates of the variables using the logit model, which affected participants' intent to purchase cherries. Multiple regression analyses showed linear results and described the estimated coefficient values as marginal effects; however, the logit model was nonlinear. Therefore, it is necessary to separate the estimated coefficient values from the marginal effects. The determinant value (Pseudo R2) of 10 explanatory variables from this model was 0.2334, and the LR (likelihood ratio) χ2 value was 43.62. The probability of χ2 (43.62) (Prob > chi2)

is 0.0007 given that the null hypothesis is true. Thus, these values like p-value indicate that the model is significant and WTP was statistically different from 0, and the model provided a better fit than the null model [45]. Therefore, the estimated expression of cherry consumers' WTP was considered a statistically appropriate model. Four of the seven dependent variables affecting participants' intent to purchase, excluding the assumed characteristics, showed significant values.

**Table 4.** Binary logistic regression for willingness to purchase.

| Category | Variable | OR | CO (SE) | z | $p > z$ | 95% CI | |
|---|---|---|---|---|---|---|---|
| Price perception | 1 | 0.36 | −1.02 (0.61) | −1.67 * | 0.09 | 0.11 | 1.19 |
| | 2 | 1.17 | 0.16 (1.44) | 0.11 | 0.91 | 0.07 | 19.57 |
| Brand familiarity | 1 | 2.65 | 0.97 (0.94) | 1.03 | 0.30 | 0.42 | 16.77 |
| | 2 | 0.28 | −1.29 (0.63) | −2.06 ** | 0.04 | 0.08 | 0.94 |
| Perceived risk | 1 | 0.86 | −0.15 (0.38) | −0.39 | 0.70 | 0.41 | 1.81 |
| | 2 | 0.38 | −0.96 (0.37) | −2.57 *** | 0.01 | 0.18 | 0.80 |
| Country of origin | Country of origin | 2.88 | 1.06 (0.53) | 2.00 ** | 0.05 | 1.02 | 8.11 |
| | Age | 1.66 | 0.51 (0.25) | 2.07 ** | 0.04 | 1.03 | 2.68 |
| | Homemaker | 0.78 | −0.25 (0.59) | −0.42 | 0.68 | 0.24 | 2.50 |
| | Married | 0.45 | −0.80 (0.73) | −1.10 | 0.27 | 0.11 | 1.87 |
| | Number of family members | 1.36 | 0.03 (0.25) | 1.20 | 0.23 | 0.82 | 2.23 |
| | Income level 1 | 2.91 | 2.11 (0.73) | 1.47 | 0.14 | 0.70 | 12.07 |
| | Income level 2 | 1.26 | 0.80 (0.64) | 0.37 | 0.71 | 0.36 | 4.39 |
| Household characteristics | Income level 3 | 1.00 | (omitted) | | | | |
| | Income level 4 | 1.00 | (omitted) | | | | |
| | Income level 5 | 1.00 | (omitted) | | | | |
| | Number of fruit purchases | 1.09 | 0.08 (0.24) | 0.35 | 0.73 | 0.68 | 1.73 |
| | Place 1 (supermarket) | 0.90 | −0.11 (0.84) | −0.13 | 0.90 | 0.17 | 4.61 |
| | Place 2 (small market) | 0.55 | −0.59 (1.13) | −0.52 | 0.60 | 0.06 | 5.10 |
| | Place 3 (traditional market) | 1.15 | 0.14 (0.95) | 0.15 | 0.88 | 0.18 | 7.39 |
| | Place 4 (online market) | 4.07 | 1.40 (1.00) | 1.40 | 0.16 | 0.57 | 29.10 |
| | Place 5 (fruit store) | 1.00 | (omitted) | | | | |

Note: OR = odds ratio, CO = coefficient, SE = standard error, and CI = confidence interval. * $p < 0.1$, ** $p < 0.05$, *** $p < 0.001$.

Among the variables that represented demographic and sociological characteristics, age ($p < 0.05$) was significant, which means that older people prefer U.S. cherries even though they have the same quality as Uzbekistan cherries and Uzbekistan has a lower price. In other words, the older one is, the more likely they were to recognize the Uzbekistan cherry market. Each value of the income level and place to purchase was converted into a dummy variable to obtain a fixed effect to obtain unbiased values. However, variables such as income, place to purchase, marital status, number of family members, and number of fruit purchases did not have significant explanatory power. Table A1 in the Appendix A provides cross-correlations across variables.

Assuming that the quality of U.S. and Uzbekistan cherries is the same and the price of Uzbekistan cherries is relatively low, consumers who perceive the current price of cherries to be high are assumed to be more willing to purchase the less expensive Uzbekistan cherries. Thus, H1 was supported.

In the case of brand family variables, consumers who tried Uzbekistan cherries were more willing to purchase Uzbekistan cherries. This indicates that consumers with cherry-eating experience have a good understanding of the quality of cherries from Uzbekistan and are more familiar with them than those without the experience, which is likely to lead to purchasing behavior. Regarding U.S. cherries, the correlation between familiarity and WTP was non-significant. This partially supported H2.

The perceived risk variables showed that consumers who have high confidence in the safety of Uzbekistan cherries were more willing to purchase them, which demonstrates that perceived risks affect purchasing behavior through direct purchasing experience or

access to relevant information. However, the correlation was non-significant between the safety assessment of U.S. cherries and consumers' WTP. This partially supported H3.

Consumers who check the country of origin are more willing to purchase U.S. cherries. Given that the country of origin is a means of verifying brand familiarity as well as product quality (safety), it indicates that preference for U.S. products is higher even if Uzbekistan ones are the same as U.S. quality and less expensive. As explained, consumers who have this tendency are highly likely to recognize the Uzbekistan cherry import market in South Korea. Thus, H4 was supported (Table 5).

**Table 5.** The results of hypotheses testing.

| Hypothesis | Result |
| --- | --- |
| **H1:** Consumers with lower (vs. higher) satisfaction with the current cherry price will be more willing to purchase less-expensive cherries from Uzbekistan | Adopted |
| **H1-1:** Consumers who value price more than quality will be more willing to purchase cherries from Uzbekistan at lower prices as compared to the U.S. | Rejected |
| **H2:** Consumers with higher (vs. lower) brand familiarity (purchasing and eating experience) will be more willing to purchase cherries from that brand | Partially adopted |
| **H3:** Consumers with a less (vs. more) perceived risk of cherry origin (U.S. or Uzbekistan) will be more willing to purchase cherries from that country. | Partially adopted |
| **H4:** Consumers who check the country of origin will be more willing to purchase cherries from advanced countries as compared to their counterparts | Adopted |

## 5. Discussion

### 5.1. Potential of Uzbekistan Cherries

Prices have a huge impact on consumers' brand decisions [12], perceived quality [24], satisfaction, and purchase intentions [25]. Between March and July, the Republic of Korea imported cherries from the United States and Uzbekistan. Korea's market share of Uzbekistan cherries is very small compared to U.S. cherries, although Uzbekistan cherries are 1.75-times less expensive (Table 1) than U.S. cherries and Uzbekistan is the world's fifth-largest exporter of cherries. Our results indicated that Korean consumers are more willing to purchase Uzbekistan cherries over U.S. cherries, especially those who are dissatisfied with the current price (H1). Since most cherries in the Korean market are U.S. cherries, the affordability of Uzbekistan cherries has a high advantage in the future.

Brand familiarity has a direct or indirect impact on purchase intention [28]. Specifically, when consumers decide to purchase, they compare product information displays with other brands. If the quality is similar, consumers will typically purchase more familiar products [29]. H2 holds that consumers with experience eating Uzbekistan cherries also have a great likelihood of purchasing Uzbekistan cherries. This suggests that Uzbekistan cherries potentially have greater market power once they increase their market share, although U.S. cherries still share a large share in the Korean cherry market.

Each decision-making process attempts to reduce risk, and risk can impact purchase decision-making if consumers are aware of the risks [30]. As expected, consumers who perceived Uzbekistan cherries as safe were willing to choose Uzbekistan cherries, which supports that potential customers may change their perception of Uzbekistan cherries from risky to safe. However, consumers who check the origin of the country preferred U.S. cherries because the halo effect of origin can have a positive effect on national image and brand familiarity [46]. Korean cherry consumers still prefer U.S. products over Uzbekistan products. Given that the stability and preference of origin increase the purchase rate when brand familiarity is low [37], U.S. cherries still prevail over Uzbekistan cherries in the market.

### 5.2. Future Implementation for More Competitive Uzbekistan Cherries

Important challenges remain for Uzbekistan cherries to be more competitive in the Korean market, such as quality improvement, price maintenance, brand marketing, and so

on. Considering the current level of Uzbekistan's fruit production and distribution base, it is unlikely that short-term traders will be activated between Korea and Uzbekistan [8]. However, the Korea Export Corporation has selected representative fruits, such as cherries, apricots, melons, and pomegranates, which are of high interest to Korean investment companies, and it aims to develop the manufacturing industry, which is expected to increase quality. Uzbekistan's cherry industry can be proposed as a model for cooperation between the two countries through official development assistance (ODA). For example, Aid for Trade (AfT) is one of the ODA initiatives that distribute the benefits of trade more equitably across and within developing countries [47]. It was launched at the 2005 Hong Kong World Trade Organization Ministerial Conference. AfT is a channel for developing countries to strengthen their competitiveness in global markets. There are five types of AfT activities: trade-related adjustment, building productive capacity, technical assistance for trade policy and regulations, trade-related infrastructure, and other trade-related needs [48]. AfT activities can upgrade the value chain, promote fair trade, and eliminate poverty in developing countries. Through AfT projects, Uzbekistan can hold a competitive position in the Korean cherry market.

## 6. Conclusions

The Korean cherry market has grown rapidly since 2000, and the demand for cherries is expected to continue to grow. U.S. cherries occupy most of the market even though Uzbekistan cherries are 1.75-times less expensive than U.S. cherries. This monopolized market has likely made the Korean cherry market unsustainable as only one provider supplies the entire set with possible side effects, including no sustainable price vector [49] and inefficiency of production and allocation [3].

By investigating WTP, this research analyzed the potential of Uzbekistan cherries to be competitive against U.S. cherries and the reason the Uzbekistan cherry market share is low. First, consumers who consider the price to be higher than the value of the product tend to prefer less-expensive products. Therefore, price competitiveness must be maintained even if the quality is improved. In particular, since products of developing countries are highly likely to be devalued, a differentiated strategy targeting the market of low-and mid-priced products would be more effective than competing with products of developed countries by increasing quality. In particular, because developing countries' products are likely to be devalued, a differentiated strategy to target the low-and mid-priced products' market may be more effective than competing with developed countries' products by increasing quality.

Second, brand familiarity is an essential factor that boosts WTP. There is a big difference in evaluation between consumers who have used or eaten the product and those who have not. Of course, if the quality of the goods does not meet consumers' expectations, any monopoly only becomes stronger. Therefore, improving consumer awareness through quality improvement and increasing their familiarity with the product through aggressive advertisement and promotion will gradually weaken the monopoly and increase consumers' preference for goods.

Third, foods such as cherries directly impact consumers' WTP because their safety is directly related to the health and life of consumers. Chinese agricultural products or food formed a monopolized market in South Korea because consumers were aware of the dangers of several food safety accidents. It takes a lot of time and cost to mitigate perceived risks. Thus, exporters need to increase their confidence in food safety for consumers in importing countries.

In conclusion, hardware-oriented support to developing countries, and support within exporting countries, such as AfT programs, is also important, and support for policy development and capacity building to establish marketing strategies, enhance safety, and support eliminating monopolized market importing countries should be combined.

This research covered only three regions of South Korea—Seoul, Incheon, and Gyeonggi-do—which limits the generalizability of our findings. Future studies should be based on

surveys of consumers in countries where monopoly markets exist. This could lead to more generalized conclusions regarding consumer preferences and purchasing patterns. Moreover, a lower level of significance concerning certain price perception, brand familiarity, and perceived risk items indicates the need for further in-depth research. Researchers should also employ a structural equation model, which can measure the indirect impact of factors on consumers' WTP in the Korean cherry market.

Despite these limitations, the results provide useful information for exporters, importers, researchers, decision-makers, and policymakers, as they show the potential for a sustainable market in the monopoly.

**Author Contributions:** Conceptualization, S.S. and S.J.; methodology, S.S. and S.J.; software, S.S.; validation, S.S. and S.J.; writing—original draft preparation, S.S.; writing—review and editing, S.S. and S.J.; visualization, S.S.; supervision, S.J.; project administration, S.J. Both authors have read and agreed to the published version of the manuscript.

**Funding:** This project was supported by the Research Resettlement Fund for the new faculty of the Institute of Green Bio Science and Technology of Seoul National University, grant number: 1403-20180082.

**Institutional Review Board Statement:** The study was conducted according to the guidelines of the Declaration of Helsinki and approved by the Seoul National University Institutional Review Board (SNUIRB) (1909/001-002, 29 August 2019).

**Informed Consent Statement:** Informed consent was obtained from all subjects involved in the study.

**Data Availability Statement:** The data that support the findings of this study are available from the corresponding author upon reasonable request.

**Acknowledgments:** This study was modified and developed from the paper presented at the 2nd International Allied Trade Associations Annual Meeting Jointly with the 23rd IAGBT-KITRI Biannual Conference and Research Symposium, held in Vietnam on 20 December 2019.

**Conflicts of Interest:** The authors declare no conflict of interest. The funder had no role in the design of the study; in the collection, analyses, or interpretation of data; in the writing of the manuscript; or in the decision to publish the results.

## Appendix A

**Table A1.** Correlation Matrices

|  | **WTP** | **PP1** | **PP2** | **BF1** | **BF2** | **PR1** | **PR2** | **CO** |
|---|---|---|---|---|---|---|---|---|
| WTP | 1.000 | | | | | | | |
| PP1 | −0.071 | 1.000 | | | | | | |
| PP2 | 0.010 | 0.0906 * | 1.000 | | | | | |
| BF1 | 0.052 | −0.031 | −0.007 | 1.000 | | | | |
| BF2 | −0.1401 * | 0.074 | 0.031 | 0.066 | 1.000 | | | |
| PR1 | −0.1212 * | 0.007 | −0.016 | −0.039 | −0.050 | 1.000 | | |
| PR2 | −0.1800 * | −0.055 | −0.015 | 0.026 | −0.021 | 0.5776 * | 1.000 | |
| CO | 0.1228 * | −0.002 | −0.1126 * | −0.088 | −0.2964 * | 0.061 | 0.013 | 1.000 |

WTP: willingness to purchase, PP: price perception, BF: brand familiarity, PR: perceived risk, CO: country of origin. * $p < 0.10$.

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
