# Peer review of "Consumers’ Willingness to Purchase Imported Cherries towards Sustainable Market: Evidence from the Republic of Korea"

_sustainability, doi:10.3390/su13105420_

Round 1
Reviewer 1 Report
The article analyses the consumer behaviour on the seasonal market of cherries, influenced by globalisation. The presented results show that this market is sensitive to elements such as: brand familiriaty, perceived risk, country of origin, consumer perception related to price. Authors conclude that consumer behaviour is sensitive to price components, experience and security, represented by the variables above, and the research aims to be a study for exporters and importers of products on a monopolised market.
The research is well argued, has objectives and hypotheses. The hypotheses have been tested, hypothesis 1 and 4 have been accepted, hypothesis 2 and 3 have been partially accepted. Research methodology is done on a sample questionnaire given to 584 respondents in Korea.
The logarithmic model should be tested statistically and the data should be presented at least at the level of the ANOVA test and of the determination coefficient resulted from the statistical testing.
The results of the study reflect the sensitivity of the request related to the mentioned elements, are adequate for the study and reflect the research. The conclusions are relevant, they reflect the research realised in three regions in Korea, that demonstrate the specific characteristics of the market.
A more thorough look into sustainability of the studied market is recommended.
Author Response
Dear Reviewer,
Many thanks for giving me the opportunity to submit a revised draft of my manuscript titled “Consumers’ Willingness to Purchase Imported Cherries towards Sustainable Market: Evidence from the Republic of Korea” to Sustainability. We very appreciate the time and effort that you and the reviewers have dedicated to providing your valuable feedback on my manuscript. We are grateful to the reviewers for their insightful comments on my paper. We have been able to incorporate changes to reflect most of the suggestions provided by the reviewers. We have highlighted the changes within the manuscript (Please find the attachment, the revised manuscript).
Here is a point-by-point response to the reviewers’ comments and concerns.
- Comment 1: The logarithmic model should be tested statistically1 and the data should be resented at least at the level of the ANOVA test2 and of the determination coefficient3 resulted from the statistical testing.
- Response: Thank you for pointing this out and we fully agree with this comment. Therefore, we have accordingly your suggestion and throughout revised the manuscript as follows:
- For the more statistical test of our logistic model, we modified Table 4 by adding coefficient and standard error.
- Second, analysis of variance (ANOVA) and t-test were conducted and added the values on Table 3 to represent our data.
- For the determination coefficient, we verify our model is significant by presenting several factors: the determinant value (Pseudo R2) of 10 explanatory variables from this model was 0.2334, but Peseudo R2 is different to R-squared that is found in OLS regression. In this regard, we used The Likelihood-Ratio test (sometimes called the likelihood-ratio chi-squared test) is a hypothesis test that helps choose the “best” model. The LR (likelihood ratio) χ2 value was 43.62. The probability of χ2 (43.62) (Prob > chi2) is 0.0007 given that the null hypothesis is true. Thus, these values like p-value indicate that the model is significant and WTP was statistically different from 0, and the model provided a better fit than the null model.
- Comment 2: A more thorough look into sustainability of the studied market is recommended.
- Your great suggestion has been interesting us to explore a new aspect. We found that regulated markets (the studied market) have several negative impacts against sustainability on the market, such as higher prices in spite of cost advantages, allocative inefficiency, pro-duction inefficiency, supernormal profit, lack of choice, and less innovation than in a competitive market. We added this point and the reasons in our Introduction (Please find the attachment, the revised manuscript).
In addition to the above comments, all spelling and grammatical errors pointed out by the reviewers have been corrected.
We look forward to hearing from you regarding our submission and to respond to any further questions and comments you may have.
Sincerely,
Reviewer 2 Report
The paper is very intersting and well-done. The methology is quite clear and precise. I suggest no other modification.
Author Response
Dear Reviewer,
Many thanks for giving me the opportunity to submit a revised draft of my manuscript titled “Consumers’ Willingness to Purchase Imported Cherries towards Sustainable Market: Evidence from the Republic of Korea” to Sustainability. We much appreciate the time and effort that you and the reviewers have dedicated to providing your valuable feedback on my manuscript. We are grateful to the reviewers for their insightful comments on my paper.
Although you mentioned "The paper is very intersting and well-done. The methology is quite clear and precise. I suggest no other modification.", we have been able to incorporate changes to make our manuscript complete. We have highlighted the changes within the manuscript (Please find the attachment, the revised manuscript).
- For the more statistical test of our logistic model, we modified Table 4 by adding coefficient and standard error.
- Second, analysis of variance (ANOVA) and t-test were conducted, and added the values on Table 3 to represent our data.
- For the determination coefficient, we verify our model is significant by presenting several factors: the determinant value (Pseudo R2) of 10 explanatory variables from this model was 0.2334, but Peseudo R2 is different to R-squared that is found in OLS regression. In this regard, we used The Likelihood-Ratio test (sometimes called the likelihood-ratio chi-squared test) is a hypothesis test that helps choose the “best” model. The LR (likelihood ratio) χ2 value was 43.62. The probability of χ2 (43.62) (Prob > chi2) is 0.0007 given that the null hypothesis is true. Thus, these values like p-value indicate that the model is significant and WTP was statistically different from 0, and the model provided a better fit than the null model.
- We found that regulated markets (the studied market) have several negative impacts against sustainability on the market, such as higher prices in spite of cost advantages, allocative inefficiency, pro-duction inefficiency, supernormal profit, lack of choice, and less innovation than in a competitive market. We added this point and the reasons in our Introduction (Please find the attachment, the revised manuscript).
In addition, all spelling and grammatical errors pointed out by the reviewers have been corrected.
We look forward to hearing from you regarding our submission and to respond to any further questions and comments you may have.
Sincerely,
Round 2
Reviewer 1 Report
I appreciate the authors' effort to improve the article and I agree with its publication in its present form.